# Lowering Barriers to Health Risk Assessments in Promoting Personalized Health Management

**DOI:** 10.3390/jpm14030316

**Published:** 2024-03-18

**Authors:** Hayoung Park, Se Young Jung, Min Kyu Han, Yeonhoon Jang, Yeo Rae Moon, Taewook Kim, Soo-Yong Shin, Hee Hwang

**Affiliations:** 1KakaoHealthCare Corp., Seongnam-si 13529, Gyeonggi-do, Republic of Korea; hpark.park@kakaohealthcare.com (H.P.); saylor.moon@kakaohealthcare.com (Y.R.M.); ray0601@snu.ac.kr (T.K.); sean.shin@kakaohealthcare.com (S.-Y.S.); drhwang.spike@kakaohealthcare.com (H.H.); 2Department of Digital Healthcare, Seoul National University Bundang Hospital, Seongnam-si 13620, Gyeonggi-do, Republic of Korea; syjung@snubh.org (S.Y.J.); jyhoon94@snu.ac.kr (Y.J.)

**Keywords:** health risk assessment, health risk prediction, personalized health management, health promotion, machine learning

## Abstract

This study investigates the feasibility of accurately predicting adverse health events without relying on costly data acquisition methods, such as laboratory tests, in the era of shifting healthcare paradigms towards community-based health promotion and personalized preventive healthcare through individual health risk assessments (HRAs). We assessed the incremental predictive value of four categories of predictor variables—demographic, lifestyle and family history, personal health device, and laboratory data—organized by data acquisition costs in the prediction of the risks of mortality and five chronic diseases. Machine learning methodologies were employed to develop risk prediction models, assess their predictive performance, and determine feature importance. Using data from the National Sample Cohort of the Korean National Health Insurance Service (NHIS), which includes eligibility, medical check-up, healthcare utilization, and mortality data from 2002 to 2019, our study involved 425,148 NHIS members who underwent medical check-ups between 2009 and 2012. Models using demographic, lifestyle, family history, and personal health device data, with or without laboratory data, showed comparable performance. A feature importance analysis in models excluding laboratory data highlighted modifiable lifestyle factors, which are a superior set of variables for developing health guidelines. Our findings support the practicality of precise HRAs using demographic, lifestyle, family history, and personal health device data. This approach addresses HRA barriers, particularly for healthy individuals, by eliminating the need for costly and inconvenient laboratory data collection, advancing accessible preventive health management strategies.

## 1. Introduction

Recent advancements in biomedicine and information technology have catalyzed a paradigm shift in healthcare, moving from treating the sick in healthcare facilities to preventing illness in healthy individuals through personalized health management in communities, a concept central to P4 (predictive, preventive, personalized, and participatory) medicine [1]. This approach, emphasizing prediction, prevention, personalization, and participation, aims to preemptively identify disease susceptibility and prevent progression through tailored healthcare interventions [2]. The success of P4 medicine increasingly relies on precise health risk assessments (HRAs), leveraging data science, wearable technology, and the Internet of Things (IoT) to predict individual health risks and potential mortality [2,3].

Originally developed in the late 1940s and evolving significantly since the mid-2000s, the applications of HRAs have transitioned from clinical settings to community health promotion programs [4,5,6,7,8,9], involving questionnaires on demographic details, lifestyle factors, medical history, and physiological data to gauge individual health risks [10,11]. Nonetheless, evidence substantiating the predictive accuracy of HRA instruments has remained limited, a situation largely attributable to the scarcity of data linking assessment inputs to health outcomes over extended time frames with regard to issues of data linkage, not to mention the imperative need for cost-effectiveness to rationalize data collection efforts vis-à-vis prediction precision, thereby expanding the instruments’ utility [12].

HRA constitutes a systematic process involving the evaluation of an individual’s health risks based on factors including lifestyle, medical history, and biomarkers [13]. However, the challenge of obtaining these data, especially from healthy individuals, is significant, as evidenced by low participation rates in wellness programs, like the 24% participation in the Annual Wellness Visit by Medicare fee-for-service beneficiaries in 2017 [14].

The objective of this study was to investigate the feasibility of conducting HRAs without relying on high-cost data such as laboratory tests, which often necessitate visits to healthcare facilities. By leveraging machine learning methods, the predictive performance of HRA models with and without laboratory data was compared and the feature importance of the models was analyzed to gain insights useful for developing personalized health management guidelines. The study results indicated that the predictive performances of the models utilizing demographic, lifestyle, family history, and personal health device data, with or without laboratory data, were comparable. Moreover, the models without laboratory data identified important features that were more valuable in developing health guidelines, thus emphasizing modifiable lifestyle factors. These findings could facilitate easier access to personalized health management for healthy individuals, thereby supporting the broader implementation of P4 medicine.

## 2. Materials and Methods

### 2.1. Data

Our study utilized the National Health Insurance Service (NHIS)-National Sample Cohort (NSC) provided by the NHIS of Korea, covering nearly all residents except those under a medical aid program funded by general taxation [15]. The NHIS-NSC, a population-based cohort, integrates four key datasets: insurance eligibility, medical check-ups, insurance claims, and death registry data. This cohort represents a 2.2% sample (1 million individuals) of NHIS members from 2002 to 2003, carefully stratified to mirror age, sex, and income distributions in Korea, with data updated through 2019. Our research primarily utilized medical check-up data (2009–2015), insurance claims data (2002–2019), and death registry data (2009–2019) from this cohort.

The NHIS administers biennial medical check-ups for beneficiaries aged 40 and above through the National Health Screening Program (NHSP). This program also includes younger blue-collar workers and household heads. Those in high-risk work environments are eligible for annual check-ups. The NHSP involves laboratory tests and self-reported health behavior and medical history questionnaires.

The insurance claims dataset, processed by the Health Insurance Review and Assessment Service (HIRA), includes details on patient identification, provider information, service descriptions, diagnoses (ICD-10 codes), and total charges. The death registry dataset, sourced from Statistics Korea [16], records the date and cause of death. We excluded deaths due to external causes such as accidents or suicides from our study, in line with Kwon et al.’s criteria [17].

Our initial dataset comprised 489,461 records from the cohort that had medical check-ups conducted between 2009 and 2012. After applying various exclusion criteria, the final dataset included 425,148 records. Exclusions were made for individuals under 30 years old (as the NHSP primarily targets adults over 40), records with character values in birth year fields, missing data, and records with extreme values indicating probable typographical errors. Figure 1 illustrates the schematic diagram of the study dataset.

### 2.2. Variables

In this study, we evaluated the health risk of individuals by quantifying the likelihood of future adverse health events, such as mortality and chronic diseases, within a predefined time frame. We accomplished this by utilizing machine learning models to predict the incidence of these events.

We categorized predictor variables into five distinct groups: demographic variables (DEMO), lifestyle variables encompassing health behaviors and body measurements (LS), family history variables (FH), personal health device variables (PHD), and laboratory variables (LAB). In Table 1, we present the predictor variables’ definitions, notations and descriptions, as well as the descriptive statistics and frequency distributions for both male and female datasets.

In recent studies, lifestyle variables, which hold the potential guiding personal lifestyle interventions to prevent or treat adverse health events, encompass multiple interconnected aspects such as body weight, body mass index (BMI), and waist circumference [18,19]. Previous studies addressing the global burden of disease have considered smoking, alcohol intake, and substance use as behavioral risk factors in efforts to mitigate health-related losses [20]. Our study incorporates lifestyle variables, including body measurements and health behavior variables; the former are measured during medical check-ups and the latter are derived from self-reported survey responses to NHIS-NSC questionnaires administered during medical check-ups.

We defined health behavior variables across three domains: smoking, alcohol intake, and physical activity; all of these directly influence an individual’s health status [21]. Smoking amount (SMK) is quantified as the cumulative amount of smoking undertaken over an individual’s lifetime in pack-years using Equation (1) [22]. Alcohol intake (DRK) is calculated as the amount of alcohol consumped weekly in bottles using Equation (2) [23]. Physical activity (PA) is computed based on parameters from NHIS-NSC questionnaires, considering light activity, moderate activity, and vigorous activity, and converting them into metabolic equivalents (METs) using Equation (3) [24].
SMK (Pack-year) = # of cigarettes a day × 0.05 × # of years smoked(1)
DRK (Bottle/week) = Mean alcohol intake a day (g) × 0.02 × # of days drank a week(2)
PA (Metabolic equivalents) = # of light activity days a week × 2.9 × 30 + # of moderate activity days a week × 4 × 30 + # of vigorous activity days a week × 7 × 20(3)

Family history variables are represented as (0, 1) indicator variables across four areas of adverse health events: heart disease, stroke, hypertension, and diabetes. These variables are computed using self-reported survey responses obtained during medical check-ups.

With the proliferation of technology and the increased accessibility of medical wearable devices, a growing reservoir of clinical data is now available outside traditional clinical settings. Such data are employed by both patients and healthy individuals to manage their health from the comfort of their homes. We refer to this subset of variables as personal health device variables (PHD), and blood pressure (BP) and fasting blood sugar (FBS) were included in this study. Our study utilizes medical check-up data to compute PHD variables.

On the other hand, we define a category for laboratory variables (LAB), encompassing measurements obtained from blood and urine samples analyzed in clinical laboratories. This includes biomarkers such as cholesterol, aspartate aminotransferase (AST), and hemoglobin (HGB). Data from medical check-ups were employed to calculate the LAB variables used in our study.

Our study focused on predicting mortality and the incidence of five major chronic diseases: heart disease, stroke, cancer, hypertension, and diabetes. These diseases are prominent contributors to global morbidity, disability, and mortality, and pose substantial individual and socioeconomic burdens due to their prolonged management and associated costs [25,26]. By predicting these adverse health events, our models aim to facilitate early detection and personalized risk management, thereby improving public health outcomes and healthcare system cost-effectiveness [27].

The study dataset, compiled from medical check-up data (2009–2012) and claims data (2002–2019), was analyzed to identify these adverse health events. Our approach involved assessing three prediction timeframes, namely three, five, and ten years, starting from the year following the medical check-up. This analysis prioritized data free from recorded health issues up to the year of the check-up, excluding records of adverse health events in or before the check-up year and those indicating mortality within the prediction timeframe.

Chronic disease incidences were determined based on the ICD-10 diagnosis codes present in the claims data and laboratory test outcomes obtained from medical check-ups. Heart disease was identified when ICD-10 codes I20–I25 were recorded as a principal or a secondary diagnosis in the claims data, and ICD-10 codes I60–I69 were associated with stroke, as used in prior studies [28,29,30,31]. Similarly, cancer incidences were detected based on principal or secondary diagnoses in the claims data, focusing on the five most common cancer types by gender [32]: lung (C33, C34), gastric (C16), colorectal (C18, C19, C20), prostate (C61), and liver (C22) cancer for male, and breast (C50, D05), colorectal (C18, C19, C20), gastric (C16), lung (C33, C34), and liver (C22) cancer for female. Thyroid cancer was excluded from the list of adverse health events in this study because the five-year survival rate in Korea is over 99% [33]. The incidences of hypertension and diabetes were established when BP ≥ 140/90 mmHg was recorded in medical check-ups or ICD-10 codes I10–I15 were recorded as a principal or a secondary diagnosis in the claims data during the data search period, and when diabetes with fasting glucose ≥ 126 mg/dL was recorded in the medical check-ups or when ICD-10 codes R81, E10–E14 were recorded as a principal or a secondary diagnosis in the claims data. Details of the number of records and the prevalence of adverse health events in the male and female datasets are presented in Table 2.

### 2.3. Analytical Models

We designed this study to evaluate whether including predictor variables with higher acquisition costs improves the predictive accuracy of HRAs for the personalized prediction and prevention of adverse health events. We systematically introduced groups of variables one at a time (Models 1–4 in Figure 1) to assess the incremental predictive accuracy gained by adding the groups of variables to the models. Conceptually, the data acquisition costs reflect the financial and logistical burden associated with obtaining the data, as well as the discomfort and inconvenience experienced by individuals during the acquisition process. We posited that acquiring laboratory data would be the most resource-intensive and cumbersome process due to the need for individuals to undergo procedures involving needles and blood extraction [34]. Considering the significantly different characteristics between male and female datasets (Table 1 and Table 2), we conducted separate analyses for each gender.

A comparative analysis of the models enabled us to examine the incremental predictive accuracy introduced by each group of variables. The models were trained on 70% of the dataset and tested on the remaining 30%. The evaluation metrics included the area under the curve (AUC), accuracy, and F1-score [35]. We utilized Youden’s J statistic to determine the optimal threshold for maximizing the accuracy and F1-score. The significance levels of the AUC differences for each model were assessed using DeLong’s method [36,37].

Our primary analytical tool was the XGBoost model, known for its exceptional predictive capabilities [38,39,40,41,42]. To validate the XGBoost results, we also applied logistic regression with stepwise variable selection. Hyperparameter optimization was performed using the grid search method [43,44,45], and multiple hyperparameter combinations were evaluated to compare their predictive performance [46]. We enhanced this process through 10-fold cross-validation. To evaluate the importance of each predictor variable, we conducted a gain analysis using XGBoost’s feature importance algorithm. All computations were performed in R version 4.3.0.

## 3. Results

Table 1 presents the categories and definitions of the predictor variables, along with the descriptive statistics for continuous variables and the frequency distributions for binary categorical variables, for both males and females. All differences in the statistics between the male and female data were statistically significant at a significance level of α = 1%. The average age for male records was 48.8, while for female records, it was 51.7. Table 2 presents the number of records used in the prediction models and the prevalence of adverse health events for each prediction timeframe, with the significance levels of the differences in prevalence between the male and female data. The lowest prevalence was observed for cancer in the three-year prediction period (1.09% for males and 0.62% for females), while the highest prevalence was noticed for hypertension in the ten-year prediction period (28.02% for males and 23.57% for females).

### 3.1. Incremental Predictive Performance Achieved by the Inclusion of Groups of Predictor Variables

Our study presents an in-depth analysis of the predictive efficacy of four models (Models 1–4), as detailed in Figure 2 and Table A1 in Appendix A. We evaluated these models based on their area under the curve (AUC), accuracy, and F1-score in the testing datasets. To assess the impact of incorporating different groups of predictor variables, we measured changes in the model performance before and after their addition. DeLong’s test results for the significance of AUC differences are provided in Table A1, with logistic regression results for comparison in Table A2 in Appendix A.

The AUC, a measure of a model’s ability to distinguish between records with and without adverse health event incidences, showed a range of 0.623 (three-year hypertension prediction for males, Model 1) to 0.897 (five-year mortality prediction for males, Model 4). Models 3 and 4 consistently achieved AUCs above 0.7 in all predictions except for female cancer predictions. Accuracy, representing the percentage of correct predictions, varied notably from 0.482 (ten-year cancer prediction for females, Model 4) to 0.830 (ten-year mortality prediction for males, Model 4). The F1-score, indicating the balance between precision and recall, ranged from a low of 0.020 (three-year cancer prediction for females, Model 1) to a high of 0.533 (ten-year hypertension prediction for males, Model 4). An interesting pattern observed in Figure 2 is the improvement in F1-scores with longer prediction timeframes, especially evident in the hypertension and diabetes predictions, as opposed to mortality and cancer.

Upon adding LS and FH variables to Model 2, the AUC values improved for most adverse health event predictions, especially for hypertension and diabetes, demonstrating the value of these variables in enhancing the prediction accuracy. Model 3, which incorporated PHD variables alongside DEMO, LS, and FH variables, showed an increase in AUC values across most predictions, with marked improvements in hypertension and diabetes predictions for both genders. However, the transition from Model 3 to Model 4, involving the addition of LAB variables, resulted in relatively modest improvements in AUC values, with limited gains in accuracy and F1-scores. These findings suggest that the inclusion of LAB variables, despite their high acquisition cost, contributed only marginally to the overall predictive performance for most health risks.

In summary, our analysis demonstrates that while the addition of LS, FH, and PHD variables significantly enhanced the predictive efficacy of our models, the incremental gain from incorporating LAB variables was limited, indicating a nuanced balance between data acquisition costs and the predictive performance in health risk assessments.

### 3.2. Feature Importance

Table 3 presents the top five significant features in our prediction models, highlighting the proportion of variance each feature explained. This analysis is crucial for developing personalized health promotion guidelines based on individual HRA results. Notably, we focused on the impact of introducing laboratory variables by comparing the key features in Model 3 (without LAB variables) and Model 4 (with LAB variables).

AGE consistently emerges as a dominant feature in most predictions. However, other variables such as SBP and FBS were more significant in predicting hypertension and diabetes, respectively. Body measurements like WC and BMI were prominent predictors for heart disease and stroke, while WT was significant for mortality. Health behavior variables like SMK (for heart disease), PA (for females in Model 2), and DRK (for three-year female cancer) were notable predictors for certain health risks. LAB variables showed varying levels of significance, with their overall contribution to predictive performance being limited when combined with other variable groups. Family history variables consistently appeared among the top five features predicting various health risks, albeit with lower rankings.

## 4. Discussion

In our study, we categorized the predictor variables of Health Risk Assessment (HRA) models into four tiers based on acquisition costs: demographic (DEMO), lifestyle (LS) and family history (FH), personal health device (PHD), and laboratory (LAB) variables. This categorization enabled us to evaluate the incremental predictive performance of each tier, balancing data acquisition costs against the predictive effectiveness of HRA models.

Our results demonstrated that Model 3, incorporating DEMO, LS, FH, and PHD variables, had a predictive performance comparable to Model 4, which added LAB variables, across various adverse health events and prediction timeframes. Interestingly, even Model 2, which included only DEMO, LS, and FH variables, performed effectively in most predictions. However, the accuracy measures, especially for stroke predictions in females, tended to decrease with the addition of PHD and LAB variables. These findings suggest that excluding costly and inconvenient LAB variables from HRAs does not significantly impair the predictive efficacy, potentially enhancing the accessibility and widespread adoption of personalized healthcare.

Feature importance analyses reinforced the well-established connections between health behavior and outcomes [20,47,48]. Notably, the significant features from Models 2 and 3, encompassing modifiable factors like WC and BMI, provided valuable insights for health guideline development compared to those from Model 4. The associations between PHD variables (SBP, DBP, FBS) and LS variables imply that lifestyle changes can influence PHD variables.

All the models in our study predicted the incidences fairly accurately, with varying degrees of accuracy depending on the type of incidence and prediction timeframe. These findings highlight the effectiveness of our assessment models in formulating personalized health promotion strategies. Notably, the first and third-ranked diseases that incurred the highest expenditures for the National Health Insurance Service of Korea in 2021 were hypertension and type 2 diabetes [49]. While the predictive performance of the models for heart diseases, stroke, and cancer can be deemed decent, further improvements may be necessary depending on the assessment’s purpose.

Additionally, these study findings bear significant implications in the era of technological advancements that enable individuals to access their health data through personal health devices without visiting healthcare facilities [50]. As the availability and reliability of personal health device data continue to improve, the depth of person-generated health data will increase, offering detailed and continuous information [51,52,53]. Our study has shown that increasing the accessibility of health data from personal health devices can be a key factor in HRA, potentially replacing data from clinical settings and expanding the market potential of HRAs.

We conducted an examination of three evaluation measures, namely AUC, accuracy, and F1-score, in the context of risk predictions for six adverse health events across three different prediction timeframes. While our overall findings align with the major trends observed, we did encounter irregular results for a few specific target risks and evaluation measures, particularly the accuracy and F1-scores. These findings underscore the importance of making judicious selections when choosing an appropriate measure in the evaluation of HRA models, depending on the specific target event to be predicted and the intended use of the assessment results. For instance, F1-scores are designed to address issues related to measuring the predictive performance in imbalanced data scenarios, such as the incidences of mortality and cancer. Additionally, when the cost associated with a false negative (missing the incidence of events in the prediction) outweighs the cost of a false positive (predicting negatives as positives), it becomes evident that sensitivity and specificity should not be given equal weight in the evaluation.

This study has limitations. Firstly, the predominance of NHSP participants over 40 years old in our database may limit its generalizability across different age groups and countries. Secondly, there may be an underestimation of disease prevalence, particularly for diseases in which individuals had a disease but were insensitive to symptoms and did not seek care at healthcare facilities. This is particularly relevant for diseases such as diabetes and hypertension, which are known to affect a large proportion of individuals who are unaware of their condition [54,55,56]. Thirdly, the reliance on self-reported questionnaire data for lifestyle and family history variables introduces the potential for omission and recall errors [57,58,59]. We anticipate that this limitation will be addressed as the accessibility and accuracy of wearable and IoT data continue to improve [60,61]. Lastly, we employed body measurements and personal health device data collected in clinical settings for our analyses and assumed their equivalence when measured in non-clinical settings. We expect this assumption would not impact the study’s overall implications, as our focus was on future risk assessment rather than immediate disease diagnosis.

Future research should explore the potential of wearables and IoT data beyond blood pressure and blood sugar measurements in HRAs. These sources of data offer accuracy, automatic reporting, non-invasiveness, and continuous monitoring. Leveraging such high-quality data has the potential to significantly enhance HRAs and contribute to more effective lifestyle modification and health promotion efforts.

## 5. Conclusions

This study aimed to assess the incremental predictive performance of four tiers of predictor variables—demographic, lifestyle and family history, personal health device, and laboratory—in predicting mortality and five chronic diseases across different timeframes. Our primary goal was to strike a balance between data acquisition costs and prediction accuracy to facilitate the widespread implementation of personalized health promotion strategies through HRAs. Our research yields three significant contributions.

Firstly, our findings indicate that the addition of laboratory variables beyond demographic, lifestyle, family history, and personal health device variables did not significantly improve model performance across all examined health events. This insight suggests that removing the need for costly and inconvenient laboratory data acquisition could lower barriers to HRAs, especially for healthy individuals, thereby enhancing accessibility to personalized health promotion.

Secondly, the models incorporating lifestyle and family history variables alongside demographic variables demonstrated comparable performance to full models when assessing the risk of heart diseases, stroke, and cancer. Notably, for certain assessments like cancer in females, the inclusion of further variables resulted in decreased accuracy and F1-scores. This underscores the reliability of Model 2 when aiming to perform accurate risk assessments for these health events.

Lastly, our analysis of important features from Models 2 and 3, which include modifiable body measurements and health behavior variables, suggests their suitability for designing health guidelines compared to models incorporating laboratory data. This implies that guidance from Models 2 and 3 is more relevant and practical for health practitioners and policymakers in shaping effective personalized health management strategies.

In conclusion, our study’s findings offer valuable insights for healthcare practitioners and policymakers, aiding in the formulation of personalized health promotion strategies without significant data acquisition costs. By extending these strategies to a broader population, including those with limited access to healthcare facilities, we can foster a new era of more accessible and effective personalized health management. As we continue to navigate the delicate balance between data acquisition costs and prediction performance, we anticipate further advancements in personalized healthcare.

## Figures and Tables

**Figure 1 jpm-14-00316-f001:**
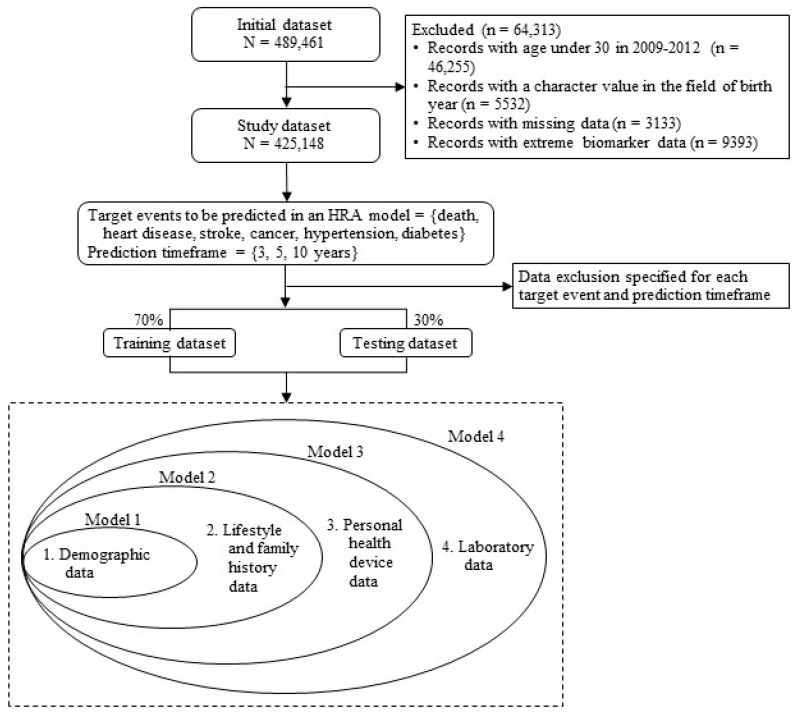
The schematic diagram of the study dataset and analytical models.

**Figure 2 jpm-14-00316-f002:**
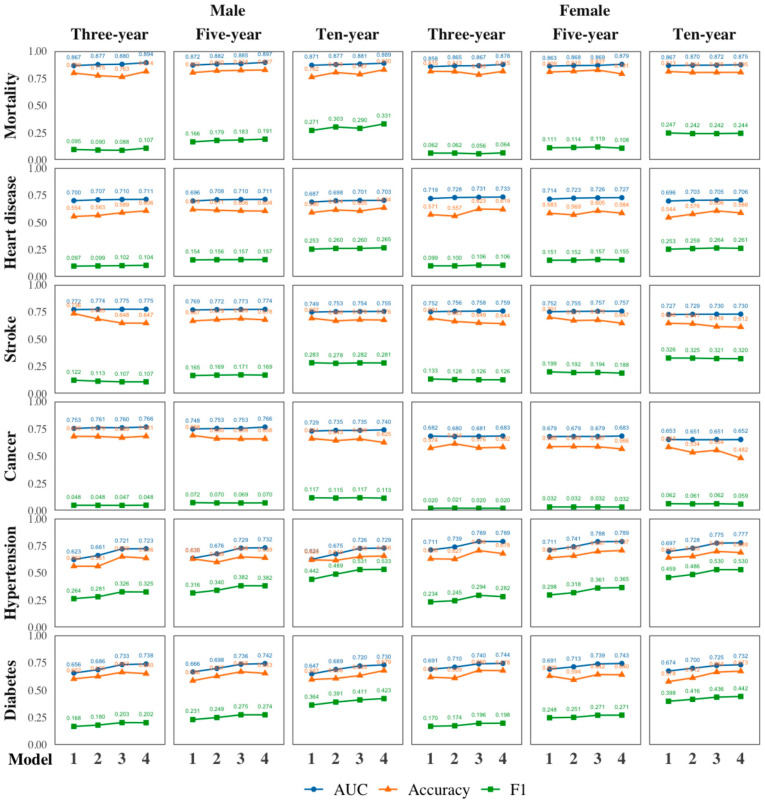
Predictive performance of the Models 1–4.

**Table 1 jpm-14-00316-t001:** Definition of predictor variables with descriptive statistics/frequency distribution for male and female datasets (N = 425,148).

Category	Variable	Definition	Mean ± STD/Freq %		*p*-Value
			Male (*n* = 214,613)	Female (*n* = 210,535)	
Demographic (DEMO)	AGE	Age (years)	48.8 ± 12.9	51.7 ± 12.6	<0.001
Health behavior (LS)	SMK	Smoking amount (pack-year)	11.9 ± 14.0	0.4 ± 2.6	<0.001
	DRK	Alcohol intake (bottle/week)	1.6 ± 2.2	0.2 ± 0.7	<0.001
	PA	Physical activity (MET-minute scores, IPAQ analysis)	540.3 ± 528.7	457.1 ± 498.0	<0.001
Body measurement (LS)	HT	Height (cm)	169.6 ± 6.4	156.1 ± 6.1	<0.001
	WT	Weight (kg)	69.9 ± 10.5	57.2 ± 8.5	<0.001
	WC	Waist circumference (cm)	83.9 ± 7.5	77.2 ± 8.7	<0.001
	BMI	Body mass index (kg/m^2^)	24.3 ± 3.0	23.5 ± 3.3	<0.001
Family history (FH)	FH_HT	Family history of heart diseases	3.4%	3.7%	<0.001
	FH_STR	Family history of stroke	6.3%	6.6%	0.004
	FH_HTN	Family history of hypertension	10.6%	13.7%	<0.001
	FH_DM	Family history of diabetes	8.8%	10.0%	<0.001
Personal health device (PHD)	SBP	Systolic blood pressure (mmHg)	125.0 ± 14.3	120.8 ± 16.0	<0.001
	DBP	Diastolic blood pressure (mmHg)	78.2 ± 9.9	74.7 ± 10.2	<0.001
	FBS	Fasting blood sugar (mg/dL)	100.4 ± 25.3	96.3 ± 21.3	<0.001
Laboratory (LAB)	TCHOL	Total cholesterol (mg/dL)	194.9 ± 35.6	197.9 ± 37.0	<0.001
	HDL	High density lipoprotein (mg/dL)	51.9 ± 12.9	57.8 ± 13.9	<0.001
	LDL	Low density lipoprotein (mg/dL)	113.6 ± 32.6	117.2 ± 33.7	<0.001
	TG	Triglycerides (mg/dL)	147.3 ± 82.4	113.6 ± 64.8	<0.001
	HGB	Hemoglobin (g/dL)	14.9 ± 1.2	12.8 ± 1.2	<0.001
	SCR	Creatinine (mg/dL)	1.0 ± 0.2	0.8 ± 0.2	<0.001
	EGFR 1	Glomerular filtration rate (GFR) ≥ 90	45.7%	48.2%	<0.001
	EGFR 2	60 ≤ GFR < 90	50.3%	46.0%	
	EGFR 3	30 ≤ GFR < 60	3.9%	5.7%	
	EGFR 4	15 ≤ GFR < 30	0.0%	0.1%	
	AST	Aspartate aminotransferase (U/L)	26.7 ± 11.7	23.1 ± 9.8	<0.001
	ALT	Alanine aminotransferase (U/L)	28.8 ± 18.5	20.2 ± 13.1	<0.001
	GGT	Gamma glutamyl transferase (U/L)	45.4 ± 37.5	22.3 ± 18.9	<0.001
	UPROT 0	Urine protein 0 g/day	94.9%	95.3%	<0.001
	UPROT 1	<0.5	2.4%	2.3%	
	UPROT 2	0.5 ≤ UPROT < 1	1.8%	1.6%	
	UPROT 3	1 ≤ UPROT < 2	0.7%	0.6%	
	UPROT 4	2 ≤ UPROT	0.2%	0.1%	

**Table 2 jpm-14-00316-t002:** Number of records (n) and prevalence (%) of adverse health events in the prediction models.

Adverse Health Event	Three Year					Five Year					Ten Year				
	Male		Female		*p*-value	Male		Female		*p*-Value	Male		Female		*p*-Value
	*n*	Prev. (%)	*n*	Prev. (%)		*n*	Prev. (%)	*n*	Prev. (%)		*n*	Prev. (%)	*n*	Prev. (%)	
Mortality	209,532	1.37	207,589	0.81	<0.001	212,522	2.48	209,740	1.52	<0.001	96,687	5.32	81,332	3.91	<0.001
Heart diseases	189,688	3.16	185,675	3.06	0.079	187,552	5.08	184,485	4.95	0.088	83,670	9.72	70,097	9.92	0.239
Stroke	197,874	2.73	191,359	3.43	<0.001	195,776	4.42	190,146	5.52	<0.001	87,349	8.86	72,347	12.24	<0.001
Cancer	205,050	1.09	204,543	0.62	<0.001	202,577	1.71	202,988	0.99	<0.001	89,590	3.16	76,757	2.12	<0.001
Hypertension	135,188	12.41	136,351	8.25	<0.001	134,262	15.48	135,939	11.41	<0.001	61,298	28.02	51,226	23.57	<0.001
Diabetes	170,187	6.27	169,365	5.77	<0.001	168,615	9.17	168,442	9.15	0.900	76,231	18.01	64,225	19.37	<0.001

**Table 3 jpm-14-00316-t003:** The top five important features and the proportion of model variance explained by each feature.

Adverse Health Event	Gender	Fea. Rank.	Three Year						Five Year						Ten Year					
			Model 2		Model 3		Model 4		Model 2		Model 3		Model 4		Model 2		Model 3		Model 4	
Mortality	Male	1	AGE	(0.816)	AGE	(0.780)	AGE	(0.646)	AGE	(0.863)	AGE	(0.832)	AGE	(0.729)	AGE	(0.910)	AGE	(0.850)	AGE	(0.769)
		2	WT	(0.056)	WT	(0.051)	HGB	(0.050)	WT	(0.046)	WT	(0.043)	HGB	(0.045)	SMK	(0.022)	WT	(0.030)	HGB	(0.027)
		3	BMI	(0.036)	BMI	(0.032)	AST	(0.035)	SMK	(0.024)	FBS	(0.028)	GGT	(0.034)	WT	(0.022)	FBS	(0.025)	GGT	(0.024)
		4	PA	(0.026)	FBS	(0.030)	GGT	(0.032)	BMI	(0.022)	SMK	(0.023)	WT	(0.029)	BMI	(0.021)	SMK	(0.022)	WT	(0.023)
		5	SMK	(0.024)	PA	(0.022)	WT	(0.031)	PA	(0.019)	BMI	(0.021)	FBS	(0.020)	PA	(0.010)	BMI	(0.020)	AST	(0.019)
	Female	1	AGE	(0.875)	AGE	(0.840)	AGE	(0.723)	AGE	(0.906)	AGE	(0.873)	AGE	(0.706)	AGE	(0.853)	AGE	(0.858)	AGE	(0.776)
		2	WT	(0.036)	WT	(0.035)	HGB	(0.051)	WT	(0.025)	WT	(0.025)	HGB	(0.039)	BMI	(0.039)	FBS	(0.030)	HGB	(0.020)
		3	BMI	(0.030)	FBS	(0.029)	GGT	(0.046)	BMI	(0.021)	FBS	(0.023)	GGT	(0.038)	WT	(0.025)	WT	(0.024)	FBS	(0.018)
		4	PA	(0.027)	BMI	(0.026)	WT	(0.030)	PA	(0.016)	HT	(0.016)	FBS	(0.024)	WC	(0.024)	BMI	(0.019)	GGT	(0.018)
		5	WC	(0.010)	PA	(0.025)	FBS	(0.024)	WC	(0.013)	DBP	(0.015)	WT	(0.024)	PA	(0.021)	WC	(0.014)	WT	(0.017)
Heart disease	Male	1	AGE	(0.739)	AGE	(0.683)	AGE	(0.617)	AGE	(0.869)	AGE	(0.835)	AGE	(0.819)	AGE	(0.809)	AGE	(0.767)	AGE	(0.734)
		2	BMI	(0.071)	BMI	(0.045)	WC	(0.042)	WC	(0.060)	WC	(0.047)	WC	(0.029)	WC	(0.067)	BMI	(0.037)	WC	(0.054)
		3	WC	(0.061)	SBP	(0.041)	BMI	(0.038)	BMI	(0.045)	BMI	(0.036)	FH_HT	(0.029)	BMI	(0.053)	WT	(0.033)	BMI	(0.033)
		4	PA	(0.027)	WC	(0.039)	SBP	(0.030)	SMK	(0.009)	SBP	(0.029)	BMI	(0.027)	SMK	(0.030)	FBS	(0.033)	SBP	(0.029)
		5	SMK	(0.026)	WT	(0.034)	FBS	(0.028)	FH_HT	(0.008)	FBS	(0.018)	SBP	(0.024)	HT	(0.011)	WC	(0.033)	FBS	(0.029)
	Female	1	AGE	(0.890)	AGE	(0.851)	AGE	(0.824)	AGE	(0.890)	AGE	(0.851)	AGE	(0.824)	AGE	(0.857)	AGE	(0.819)	AGE	(0.786)
		2	SMK	(0.184)	WC	(0.044)	WC	(0.045)	SMK	(0.184)	WC	(0.044)	WC	(0.045)	BMI	(0.043)	SBP	(0.035)	SBP	(0.029)
		3	WC	(0.051)	SBP	(0.039)	DBP	(0.021)	WC	(0.051)	SBP	(0.039)	DBP	(0.021)	WC	(0.037)	WC	(0.033)	BMI	(0.024)
		4	BMI	(0.021)	BMI	(0.020)	SBP	(0.021)	BMI	(0.021)	BMI	(0.020)	SBP	(0.021)	WT	(0.021)	BMI	(0.031)	WC	(0.023)
		5	WT	(0.016)	FBS	(0.014)	BMI	(0.017)	WT	(0.016)	FBS	(0.014)	BMI	(0.017)	SMK	(0.015)	FBS	(0.021)	GGT	(0.020)
Stroke	Male	1	AGE	(0.973)	AGE	(0.957)	AGE	(0.948)	AGE	(0.923)	AGE	(0.896)	AGE	(0.866)	AGE	(0.934)	AGE	(0.915)	AGE	(0.892)
		2	WC	(0.014)	SBP	(0.011)	SBP	(0.009)	WC	(0.023)	FBS	(0.019)	FBS	(0.014)	WC	(0.024)	WC	(0.014)	WC	(0.016)
		3	SMK	(0.004)	FBS	(0.008)	FBS	(0.008)	SMK	(0.011)	SBP	(0.016)	SBP	(0.013)	SMK	(0.010)	SBP	(0.011)	SBP	(0.010)
		4	BMI	(0.004)	DBP	(0.007)	WC	(0.006)	PA	(0.011)	WC	(0.015)	WC	(0.012)	DRK	(0.007)	FBS	(0.011)	FBS	(0.009)
		5	PA	(0.003)	PA	(0.005)	DBP	(0.005)	BMI	(0.011)	SMK	(0.010)	GGT	(0.010)	FH_STR	(0.007)	HT	(0.011)	DBP	(0.008)
	Female	1	AGE	(0.929)	AGE	(0.901)	AGE	(0.877)	AGE	(0.917)	AGE	(0.889)	AGE	(0.860)	AGE	(0.973)	AGE	(0.960)	AGE	(0.950)
		2	WC	(0.022)	SBP	(0.019)	SBP	(0.017)	WC	(0.021)	SBP	(0.019)	SBP	(0.014)	WC	(0.009)	SBP	(0.009)	SBP	(0.008)
		3	BMI	(0.015)	WC	(0.013)	WC	(0.011)	BMI	(0.018)	WC	(0.017)	TG	(0.013)	BMI	(0.008)	FBS	(0.008)	WC	(0.007)
		4	PA	(0.010)	FBS	(0.012)	DBP	(0.010)	PA	(0.011)	FBS	(0.015)	WC	(0.010)	PA	(0.002)	WT	(0.005)	TG	(0.006)
		5	HT	(0.008)	DBP	(0.012)	TG	(0.009)	HT	(0.010)	BMI	(0.013)	FBS	(0.010)	WT	(0.002)	DBP	(0.005)	FBS	(0.005)
Cancer	Male	1	AGE	(0.869)	AGE	(0.855)	AGE	(0.782)	AGE	(0.916)	AGE	(0.895)	AGE	(0.821)	AGE	(0.872)	AGE	(0.847)	AGE	(0.776)
		2	SMK	(0.029)	SMK	(0.026)	AST	(0.028)	SMK	(0.030)	SMK	(0.029)	LDL	(0.029)	SMK	(0.041)	SMK	(0.038)	LDL	(0.041)
		3	BMI	(0.029)	DBP	(0.019)	SMK	(0.021)	WT	(0.011)	FBS	(0.013)	AST	(0.026)	WC	(0.022)	FBS	(0.021)	SMK	(0.032)
		4	WC	(0.021)	WC	(0.017)	HGB	(0.020)	DRK	(0.011)	SBP	(0.012)	SMK	(0.025)	BMI	(0.021)	BMI	(0.017)	FBS	(0.019)
		5	WT	(0.016)	BMI	(0.016)	LDL	(0.020)	WC	(0.011)	DRK	(0.010)	TG	(0.014)	PA	(0.015)	WC	(0.017)	HGB	(0.016)
	Female	1	AGE	(0.908)	AGE	(0.882)	AGE	(0.777)	AGE	(0.962)	AGE	(0.954)	AGE	(0.886)	AGE	(0.772)	AGE	(0.711)	AGE	(0.589)
		2	DRK	(0.025)	FBS	(0.029)	AST	(0.050)	PA	(0.099)	SMK	(0.112)	AST	(0.048)	WC	(0.042)	DBP	(0.048)	LDL	(0.046)
		3	WC	(0.016)	DRK	(0.021)	ALT	(0.031)	SMK	(0.018)	FBS	(0.011)	HGB	(0.013)	PA	(0.041)	BMI	(0.040)	TG	(0.045)
		4	BMI	(0.015)	SBP	(0.015)	TCHOL	(0.026)	WC	(0.007)	SBP	(0.005)	LDL	(0.013)	BMI	(0.040)	SBP	(0.034)	AST	(0.044)
		5	SMK	(0.014)	SMK	(0.011)	HGB	(0.023)	BMI	(0.006)	PA	(0.004)	SMK	(0.012)	HT	(0.039)	WC	(0.030)	DBP	(0.030)
Hypertension	Male	1	AGE	(0.546)	SBP	(0.350)	AGE	(0.286)	AGE	(0.574)	AGE	(0.318)	AGE	(0.277)	AGE	(0.513)	SBP	(0.330)	AGE	(0.304)
		2	BMI	(0.200)	AGE	(0.299)	FBS	(0.270)	BMI	(0.191)	SBP	(0.315)	SBP	(0.215)	BMI	(0.227)	AGE	(0.312)	DBP	(0.222)
		3	WC	(0.124)	DBP	(0.165)	SBP	(0.189)	WC	(0.124)	DBP	(0.137)	DBP	(0.163)	WC	(0.139)	DBP	(0.139)	SBP	(0.170)
		4	DRK	(0.074)	WC	(0.047)	DBP	(0.056)	DRK	(0.061)	BMI	(0.053)	BMI	(0.068)	DRK	(0.057)	WC	(0.073)	BMI	(0.092)
		5	HT	(0.021)	BMI	(0.042)	WC	(0.046)	HT	(0.018)	FBS	(0.047)	FBS	(0.043)	FH_HTN	(0.019)	BMI	(0.066)	WC	(0.053)
	Female	1	AGE	(0.709)	SBP	(0.378)	SBP	(0.401)	AGE	(0.711)	SBP	(0.378)	SBP	(0.369)	AGE	(0.484)	SBP	(0.349)	SBP	(0.303)
		2	BMI	(0.161)	AGE	(0.352)	AGE	(0.345)	BMI	(0.160)	AGE	(0.366)	AGE	(0.355)	BMI	(0.192)	AGE	(0.263)	AGE	(0.229)
		3	WC	(0.060)	DBP	(0.121)	DBP	(0.097)	WC	(0.082)	DBP	(0.107)	DBP	(0.103)	WC	(0.095)	BMI	(0.086)	BMI	(0.064)
		4	HT	(0.019)	BMI	(0.051)	BMI	(0.048)	FH_HTN	(0.016)	BMI	(0.054)	BMI	(0.049)	PA	(0.070)	WC	(0.059)	DBP	(0.046)
		5	FH_HTN	(0.017)	WC	(0.041)	WC	(0.028)	HT	(0.013)	WC	(0.048)	WC	(0.043)	HT	(0.051)	DBP	(0.059)	WC	(0.043)
Diabetes	Male	1	AGE	(0.657)	FBS	(0.468)	FBS	(0.381)	AGE	(0.668)	AGE	(0.409)	AGE	(0.389)	AGE	(0.536)	AGE	(0.320)	AGE	(0.311)
		2	WC	(0.129)	AGE	(0.318)	AGE	(0.307)	WC	(0.129)	FBS	(0.363)	FBS	(0.364)	BMI	(0.185)	FBS	(0.314)	FBS	(0.244)
		3	BMI	(0.118)	WC	(0.061)	SBP	(0.054)	BMI	(0.124)	WC	(0.064)	WC	(0.052)	WC	(0.142)	BMI	(0.113)	GGT	(0.086)
		4	FH_DM	(0.028)	BMI	(0.050)	WC	(0.042)	FH_DM	(0.022)	BMI	(0.058)	GGT	(0.045)	SMK	(0.036)	WC	(0.079)	WC	(0.063)
		5	SMK	(0.027)	SMK	(0.025)	GGT	(0.040)	SMK	(0.020)	SBP	(0.056)	ALT	(0.041)	FH_DM	(0.029)	SMK	(0.033)	SBP	(0.049)
	Female	1	AGE	(0.731)	AGE	(0.500)	AGE	(0.461)	AGE	(0.739)	AGE	(0.535)	AGE	(0.494)	AGE	(0.650)	AGE	(0.463)	AGE	(0.404)
		2	BMI	(0.113)	FBS	(0.232)	FBS	(0.265)	BMI	(0.117)	FBS	(0.200)	FBS	(0.253)	BMI	(0.157)	FBS	(0.160)	FBS	(0.225)
		3	WC	(0.095)	HT	(0.102)	GGT	(0.061)	WC	(0.099)	HT	(0.090)	WC	(0.051)	WC	(0.115)	SBP	(0.122)	WC	(0.074)
		4	FH_DM	(0.018)	WC	(0.048)	WC	(0.051)	FH_DM	(0.019)	BMI	(0.045)	BMI	(0.044)	FH_DM	(0.022)	WC	(0.087)	BMI	(0.064)
		5	HT	(0.012)	BMI	(0.033)	BMI	(0.029)	HT	(0.009)	WC	(0.033)	GGT	(0.042)	PA	(0.015)	BMI	(0.058)	TG	(0.059)

Feature names: AGE: Age; ALT: Alanine aminotransferase; BMI: Body mass index; DBP: Diastolic blood pressure; DRK: Alcohol intake; EGFR: Estimated glomerular filtration rate; FBS: Fasting blood sugar; FH_DM: Family history of diabetes; FH_HTN: Family history of hypertension; GGT: Gamma glutamyl transferase; HGB: Hemoglobin; HT: Height; PA: Physical activity; SBP: Systolic blood pressure; SCR: Creatinine; SMK: Smoking amount; TCHOL: Total cholesterol; TG: Triglycerides; WC: Waist circumference; WT: Weight.

## Data Availability

The authors do not have the authority to provide access to the NHIS-NSC data to other researchers. Access to the study data is strictly regulated by the NHIS of Korea. Acquiring access entails applying for an account to log in to their server, working exclusively within their server environment, and securing their approval to retrieve results from it.

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
