# Peer review of "Lowering Barriers to Health Risk Assessments in Promoting Personalized Health Management"

_jpm, 2024, doi:10.3390/jpm14030316_

Round 1

Reviewer 1 Report

Comments and Suggestions for Authors

Dear Authors, 

Although the fact that the evaluation was made in a certain age group and on a group (one country) is a limiting aspect of the study, it can be accepted that it is important to contribute data to the literature. 

Brief information about machine learning methodologies can be given in the introduction section.

The use of machine learning methodologies in this study should be briefly mentioned and the originality it adds to this study should be emphasized.

A reference to equality (1) should be added on line 126.

A detailed study is presented. I believe that this study will contribute to the literature and shed light on new research.

Author Response

Brief information about machine learning methodologies can be given in the introduction section.

We have revised the sentence on Line 62 to explicitly mention the use of machine learning methods throughout our analyses. While we value the reviewer's suggestion to highlight the originality added by employing these methodologies in our study, after careful consideration, we have chosen not to explicitly state it.

The use of machine learning methodologies in this study should be briefly mentioned and the originality it adds to this study should be emphasized.

A reference to equality (1) should be added on line 126.

We moved the reference [22] to the equation (1) which was misplaced, and added the reference [23] to the equation (2):

23. Kim YG, Han KD, Choi JI, Boo KY, Kim DY, Lee KN, Shim J, Kim JS, Kim YH. Frequent drinking is a more important risk factor for new-onset atrial fibrillation than binge drinking: a nationwide population-based study. Europace. 2020 Feb 1;22(2):216-224. doi: 10.1093/europace/euz256

Reviewer 2 Report

Comments and Suggestions for Authors

This is an interesting and noteworthy study.

Introduction. 

1. Line 36, bio should be bioinformatics; bio is an abbreviation for biology or biomedicine but here it is undefined.

2. P4 medicine is undefined. Please state this even with parenthesis (predictive, preventive, personalized, participatory).

3. Statement ending in Line 53 needs a reference. There are multiple sources that cite the problem of linking data with health outcomes.

4. Line 61, please use passive phrasing consistently. Rephrase from "We compared..." to "Predictive performance of HRAs with and without data were compared..."

Methods

5. For consistency, please change "our" to passive format throughout, such as line 72 ("Our study..." to "This study..."), line 92 ("Our initial data set..." to "The initial...."), line 101 ("we evaluated..." to "the health risk was evaluated.."), line 105 ("we categorized" to "Predictor variables were categorized..."). There are numerous instances of this...please correct these and be consistent throughout.

6. Was this study approved by any review board or committee - either by Seoul National University or KakaoHealthCare? There's no mention of this. You must provide this information with approval numbers of the IRB or Office for the Protection of Research Subjects. You are handling sensitive patient information, therefore unless it's deidentified in some way...you must have informed consent for all data. This information is missing. You must describe this.

7. If patient data was used......how were the data handled? Who had access? And how do you protect the release of sensitive patient information?

Results

8. Same comment as above (keep in passive voice)...change line 226 from "Our study presents..." to "This study presents...". We and Our need to be changed throughout to keep this consistent throughout this manuscript.

General comment: Very good description of outcomes and data analysis here. Clear and easy to follow outcomes for Models 1 - 4 and good insight into how variables can be combined to increase predictive effectiveness.

Discussion

9. Same comment as above (keep in passive voice)...change line 12 from "in our study we..." to "This study categorized...". We and Our need to be changed throughout to keep this consistent throughout this manuscript.

Author Response

1. Line 36, bio should be bioinformatics; bio is an abbreviation for biology or biomedicine but here it is undefined.

“bio” was replaced by “biomedicine” as the reviewer suggested.

2. P4 medicine is undefined. Please state this even with parenthesis (predictive, preventive, personalized, participatory).

The acronym P4 was defined as the reviewer suggested.

3. Statement ending in Line 53 needs a reference. There are multiple sources that cite the problem of linking data with health outcomes.

We revised the sentence and added a reference below which systematically reviewed the studies of data linkage issues:

12. Elstad M, Ahmed S, Røislien J, Douiri A. Evaluation of the reported data linkage process and associated quality issues for linked routinely collected healthcare data in multimorbidity research: a systematic methodology review. BMJ Open. 2023 May 8;13(5):e069212. doi: 10.1136/bmjopen-2022-069212

4. Line 61, please use passive phrasing consistently. Rephrase from "We compared..." to "Predictive performance of HRAs with and without data were compared..."

Methods

5. For consistency, please change "our" to passive format throughout, such as line 72 ("Our study..." to "This study..."), line 92 ("Our initial data set..." to "The initial...."), line 101 ("we evaluated..." to "the health risk was evaluated.."), line 105 ("we categorized" to "Predictor variables were categorized..."). There are numerous instances of this...please correct these and be consistent throughout.

8. Same comment as above (keep in passive voice)...change line 226 from "Our study presents..." to "This study presents...". We and Our need to be changed throughout to keep this consistent throughout this manuscript.

General comment: Very good description of outcomes and data analysis here. Clear and easy to follow outcomes for Models 1 - 4 and good insight into how variables can be combined to increase predictive effectiveness.

Discussion

9. Same comment as above (keep in passive voice)...change line 12 from "in our study we..." to "This study categorized...". We and Our need to be changed throughout to keep this consistent throughout this manuscript.

We respectfully disagree with the reviewer's suggestion that the passive voice should be consistently maintained throughout the manuscript. We believe that both passive and active constructions can be utilized where appropriate, resulting in improved readability. This approach aligns with common practice in academic writing. Upon thorough review of the manuscript, we have made revisions by interchanging active and passive constructions as needed to enhance clarity and flow.

6. Was this study approved by any review board or committee - either by Seoul National University or KakaoHealthCare? There's no mention of this. You must provide this information with approval numbers of the IRB or Office for the Protection of Research Subjects. You are handling sensitive patient information, therefore unless it's deidentified in some way...you must have informed consent for all data. This information is missing. You must describe this.

7. If patient data was used......how were the data handled? Who had access? And how do you protect the release of sensitive patient information?

The reviewer's concerns have been effectively addressed in the Institutional Review Board Statement (Lines 117-119 on page 15) and the Data Availability Statement (Lines 120-123 on page 15), as well as on Lines 74-82 in the data section on page 2.